# Lenalidomide Efficacy in Patients with MDS and Del-5q: Real-World Data from the Hellenic (Greek) National Myelodysplastic & Hypoplastic Syndromes Registry (EAKMYS)

**DOI:** 10.3390/cancers17091388

**Published:** 2025-04-22

**Authors:** Argiris Symeonidis, Panagiotis Diamantopoulos, Athanasios Galanopoulos, Alexandra Kourakli, Eleni Sazakli, Eleftheria Hatzimichael, Maria Pagoni, Panagiotis Zikos, Theodoros P. Vassilakopoulos, Eleni Gavrilaki, Anthi Bouchla, Anna Kioumi, Katerina Palla, Ioannis Kotsianidis, Evridiki Michali, Zafiris Kartassis, Eirini Katodritou, Vasileios Lazaris, Maria Vagia, George Xanthopoulidis, Theodora Assimakopoulou, Charalampos Pontikoglou, Maria Dimou, Maria Dalekou-Tsolakou, Dimitra Liapi, Maria Kotsopoulou, Vassiliki Labropoulou, Menelaos Papoutselis, Despina Barmparousi, Efthymia Vlachaki, Georgia Kaiafa, Eleni Chandrinou, Panagiotis Karmas, Evangelos Terpos, George Vassilopoulos, Panayiotis Panayiotidis, Nora-Athina Viniou, Vassiliki Pappa

**Affiliations:** 1Hematology Division, Department of Internal Medicine, University of Patras, 265 04 Patras, Greece; akourakli@gmail.com (A.K.); vaslazaris@upatras.gr (V.L.); vaslabrop@upatras.gr (V.L.); 2First Department of Internal Medicine, National and Kapodistrian University of Athens, Laikon General Hospital, 115 27 Athens, Greece; diamp@med.uoa.gr (P.D.); noravi@med.uoa.gr (N.-A.V.); 3Department of Hematology, G. Gennimatas General Hospital, 115 27 Athens, Greece; agalanopoulos@euroclinic.gr (A.G.); eviamich@hotmail.com (E.M.); 4Laboratory of Public Health, University of Patras, Medical School, 265 04 Patras, Greece; elsazak@upatras.gr; 5Department of Hematology, University of Ioannina, School of Medicine, 451 10 Ioannina, Greece; ehatzim@cc.uoi.gr; 6Hematology-Lymphoma Department—BMT Unit, Evangelismos Hospital, 106 76 Athens, Greece; marianpagoni@yahoo.com (M.P.); m.g.vagia@army.gr (M.V.); 7Department of Hematology, “St Andreas” General Hospital, 263 32 Patras, Greece; pzikos@agandreashosp.gr; 8Department of Hematology, Laikon General Hospital, National and Kapodistrian University of Athens, 115 27 Athens, Greece; tvassilak@med.uoa.gr; 9Department of Hematology & Stem Cell Transplantation, G. Papanicolaou General Hospital of Thessaloniki, 570 10 Exochi, Greece; gavriiel@auth.gr; 10Second Department of Internal Medicine, University General Hospital Attikon, 124 62 Athens, Greece; ampouchla@med.uoa.gr (A.B.); vaspappa@med.uoa.gr (V.P.); 11Department of Hematology, “Papageorgiou” Hospital, 546 24 Thessaloniki, Greece; annakioumi@gmail.com; 12Department of Hematology, General Hospital of Chania, 733 00 Chania, Greece; kpalla@euroclinic.gr; 13Department of Hematology, Democritus University of Thrace Medical School, 681 00 Alexandroupolis, Greece; ikotsian@med.duth.gr (I.K.); menelaospapoutselis@yahoo.co.uk (M.P.); 14Hematology Unit, General Hospital of Halkida, 341 00 Halkida, Greece; zkartasis@gmail.com; 15Department of Hematology, Theageneion Hospital, 546 39 Thessaloniki, Greece; aimatologiki@theagenio.gov.gr; 16Department of Hematology, General Hospital of Xanthi, 671 00 Xanthi, Greece; xanthogeo99@gmail.com; 17Department of Hematology, Sismanogleion Hospital, 151 26 Athens, Greece; theass56@yahoo.gr; 18Department of Hematology, University of Crete Medical School, 710 03 Heraklion, Greece; c.pontikoglou@med.uoc.gr; 19First Propedeutic Department of Internal Medicine, National and Kapodistrian University of Athens, Laikon General Hospital, 115 27 Athens, Greece; msdimou@med.uoa.gr (M.D.); ppanayi@med.uoa.gr (P.P.); 20Hematology and Thalassemia Unit, General Hospital of Kalamata, 241 00 Antikalamos, Greece; dalekoum@nosokomeiokalamatas.gr; 21Department of Hematology, Venizeleion Hospital, 714 09 Heraklion, Greece; dliapi@venizeleio.gr; 22Department of Hematology, Metaxa Anti-Cancer Hospital, 185 37 Piraeu, Greece; m.kotsopoulou@metaxa-hospital.gr; 23Department of Hematology, Alexandra General Hospital, 115 28 Athens, Greece; barbarousi@gmail.com; 24Second Department of Internal Medicine, Hippokration Hospital, Aristotle University, 541 24 Thessaloniki, Greece; efivlachaki@auth.gr; 25First Propedeutic Department of Internal Medicine, AHEPA University Hospital, 546 36 Thessaloniki, Greece; gdkaiafa@auth.gr; 26Department of Hematology, Athens Medical Center, 151 25 Athens, Greece; e.chandrinou@iatriko.gr; 27Department of Hematology, Army Share Fund Hospital “NIMTS”, 115 24 Athens, Greece; p.a.karmas@army.gr; 28Department of Clinical Therapeutics, National & Kapodistrian University of Athens, School of Medicine, Alexandra Hospital, 106 79 Athens, Greece; eterpos@med.uoa.gr; 29Department of Hematology, University of Thessaly, School of Medicine, 413 34 Larissa, Greece; gvasilop@bioacademy.gr

**Keywords:** myelodysplastic syndromes, del(5)q, lenalidomide, treatment, real-world-data, prognosis

## Abstract

Lenalidomide has been approved for the treatment of transfusion-dependent patients with lower-risk MDS, exhibiting the del(5)q cytogenetic abnormality. Only sporadic cases, not thoroughly analyzed, describe information on lenalidomide treatment to patients with additional cytogenetic abnormalities, over del(5)q, or to patients with del(5)q and an excess of marrow blasts. Results from a harvest of 238 patients registered in the Greek National MDS Registry have shown that lenalidomide may be effective, not only in the del(5)q syndrome, as is described in previous MDS treatment recommendations and guidelines, but also in other MDS patients exhibiting del(5)q alone or with one additional cytogenetic abnormality, irrespective of BM blast cell percentage, a finding implying that in these patient groups, the dominant disease pathophysiology is governed by the presence of the del(5)q clone.

## 1. Introduction

Among patients with myelodysplastic syndromes (MDS) or Myelodysplastic Neoplasms (MDN), the most common single cytogenetic abnormality is deletion of the long arm of chromosome 5 [del(5)q], occurring in about 10–15% of patients [1]. Del(5)q can occur either isolated (10–14% of the patients) or in association with additional cytogenetic abnormalities (5–10%) [2,3]. Del(5)q is currently the only cytogenetic abnormality that defines a specific MDS subtype/clinical entity. According to the latest World Health Organization (WHO) classification, the previously described del(5)q syndrome, which demanded thorough morphologic and cytogenetic criteria to be diagnosed as a separate and distinct clinicohematological entity, now changes its precise terminology and its diagnostic criteria. The new entity is now defined as low-risk MDS-del(5)q and is established when there is clear marrow dysplasia without an excess of blasts, associated with a cytogenetic profile, which includes either an isolated del(5)q abnormality or, at most, only one additional cytogenetic abnormality, excluding monosomy 7 or del(7)q [4]. Patients with this entity are usually characterized by the presence of macrocytic anemia, mild neutropenia, and usually normal or moderately increased platelets, dysplasia in 1–3 lineages, absence of Auer rods, <2% peripheral blood blasts, and <5% bone marrow (BM) blasts [1,3]. The previously required strict morphological criteria of dysplastic, small non-lobulated megakaryocytes or other well-specified morphological features, such as the Pelger-Huet type neutrophils and the macrocytic red blood cells, are no longer necessary to establish the diagnosis of the disease. Pathogenetically, the disease is considered an acquired ribosomopathy, attributed to haploinsufficiency of genes encoding for the ribosomal protein RPS14, resulting in impairment of the appropriate ribosomal structure and function, and consequently in disturbed protein synthesis. Although quite uncommon, a small proportion of patients might also present mutations of the spliceosomal protein SF3B1, and these patients may exhibit ring sideroblasts in their marrows upon iron staining. Real-world evidence suggests that about 50–65% of patients with isolated del(5)q fulfill criteria for the del(5)q syndrome [5,6]. No published data exist to compare clinical features of the del(5)q syndrome with the newly defined entity of the WHO-2022 classification, MDS-del(5)q.

Chronic anemia is the dominant feature of this disease and places a notable burden on patients with MDS in general, impacting their quality of life and usually leading to red blood cell (RBC) transfusion dependency [1], which is a poor prognostic indicator and has been associated with shorter event-free and overall survival (OS) [7]. Patients BM exhibits variable proportions of RBC precursors, with a high-degree of apoptosis. Erythropoietin stimulating agents (ESAs), which represent the first choice of treatment for patients with lower-risk MDS in general [8], and when early administered, might delay the onset of transfusion dependency [9], in this patient group are less effective even when serum erythropoietin levels are not elevated. In patients with the del(5q) abnormality, the disruption of erythropoiesis is severe and complex, and the probability of attaining a favorable response to ESAs is substantially lower, or responses are milder and of shorter duration [10], compared to those observed in patients without this cytogenetic abnormality. However, in patients with isolated del(5)q abnormality, including those with the del(5)q syndrome, lenalidomide, a synthetic immunomodulatory thalidomide derivative, induces significantly higher erythroid response rates, exerting direct and specific cytotoxicity against the dysplastic/neoplastic clone and decreasing the risk of acute myeloid leukemia (AML) progression or death, as compared to placebo [11]. In vitro studies have shown that lenalidomide selectively inhibits the growth of clonal erythroblasts by upregulating the haplo-deficient oncosuppressor gene SRARC, as well as by down-regulating several other genes involved in erythroid differentiation, such as HBA2, GYPA, and KLF1 [12]. In another study, the addition of lenalidomide in cultures of del(5)q leukemic cells induced cell cycle arrest in the G2 phase by inhibition of two important cell cycle regulators, Cdc25C and PP2Ac phosphatases, and induction of apoptosis, findings not observed in non-del(5)q AML cells [13,14]. Additional antileukemic activity of lenalidomide, not restricted to del(5)q cells, is related to miRNAome modulation. In particular, it downregulates the expression of pro-apoptotic miR-34a and miR-34a* in normal monocytes and upregulates the haploinsufficient miR-378 and miR-378* [15]. It also upregulates the expression of tumor suppressive miR-181a, enhancing translation of the CEPB_-p30 isoform [16]. Finally, lenalidomide induces lipid raft assembly in the erythroid cell membrane, stabilizing the erythropoietin receptor and promoting Epo-signaling through the JAK-STAT pathway [17]. Thus, lenalidomide has been approved for transfusion-dependent patients with low-risk MDS-del(5)q when other treatment approaches, including ESAs, are inefficient [18]. Lenalidomide, however, is capable of inducing complete cytogenetic remission in a substantial percentage of treated MDS-del(5)q patients, restricting clonal size and restoring non-clonal hematopoiesis [19]; therefore, the initially defined terms and recommendations for its use as a second-line treatment have become disputable, in view of the evident need for earlier start of treatment and faster clonal shrinkage/elimination to prevent potentially hazardous molecular events that might accelerate disease progression [20]. 

Although now considered the standard of care for patients with isolated del(5)q [1], lenalidomide has not been widely used in patients exhibiting an excess of marrow blasts and/or additional cytogenetic abnormalities over del(5)q. Such patients, when ineligible to undergo allogeneic stem cell transplantation (Allo-SCT), are usually treated with hypomethylating agents or supportive care alone; hence, real-world experience with lenalidomide treatment in these patient groups is limited. However, early after the initial application of the drug, some unexpected responses among patients with complex karyotypes, including or not del(5)q or even with frank AML, were noted, raising the interest for a potential use of lenalidomide in higher-risk patients [21,22,23]. More recently, several case reports have described a beneficial result of lenalidomide treatment in patients with lower-risk disease, exhibiting one additional cytogenetic abnormality, on top of del(5)q [24,25,26], yet unsuccessful treatment attempts may not be described. The aim of this retrospective study of the Greek MDS Study Group was to provide real-world evidence for the efficacy and safety of lenalidomide as a single-agent treatment in patients with del(5)q syndrome, as well as in those with an isolated del(5)q cytogenetic abnormality, but not fulfilling criteria for del(5)q syndrome, and in those with del(5)q plus additional cytogenetic abnormalities.

## 2. Patients & Methods

All patients with MDS, exhibiting del(5)q, who were registered in the Hellenic (Greek) National Myelodysplastic & Hypoplastic Syndromes Registry (EAKMYS) and started lenalidomide single-agent treatment between 1 January 2006, and 30 September 2023, were included in this study on an intention-to-treat (ITT) basis, if they had received at least one dose of lenalidomide, if full and clear response assessments could be applied, and if complete follow-ups were available. Patients who completed at least two cycles of lenalidomide treatment were considered evaluable for response. However, one patient, who responded excellently following less than one complete treatment cycle (16 days) but did not continue treatment due to grade 3–4 skin toxicity, was also included in the efficacy analysis. Similar responses following short exposure to lenalidomide have previously been described in occasional patients [27,28,29]. The majority of the patients (N = 198 or 83.2%) received 10 mg lenalidomide for 21 consecutive days every 4 weeks as a starting dose. In the remaining 40 patients (16.8%), the dose of lenalidomide initially administered had to be adapted to 5 mg for 21 or 28 consecutive days, q4w, due to mild/moderate renal impairment, as this is recommended [18,30] or to patient frailty. Prognostic categorization was assessed using the International Prognostic Scoring System (IPSS) [31], the Revised IPSS (IPSS-R) [32], and the WHO classification-based Prognostic Scoring System (WPSS) [33]. 

Based on their hematological and cytogenetic findings, and according to WHO criteria [4], patients were divided into four groups: Group-I, patients with the del(5)q syndrome (N = 153); Group-II, patients exhibiting isolated del(5)q, but not fulfilling criteria for del(5)q syndrome (N = 34); Group-III, patients exhibiting del(5)q plus only one additional cytogenetic abnormality, not affecting chromosome 7 (N = 26); and Group-IV, patients exhibiting del(5)q plus more than one additional cytogenetic abnormality or one additional abnormality affecting chromosome 7 (N = 25). Regarding the single additional abnormality of patients classified to group III, in 17 of them this was manifested as an evolutionary subclone over the del(5)q abnormality, whereas in 8 patients there was an additional clone, independent of the del(5)q clone.

Outcomes were assessed in evaluable patients and in the ITT group and included hematological response, time to obtain a favorable response, duration of response, cytogenetic response, and OS. Changes from baseline in hematological parameters (hemoglobin [Hb], white blood cell [WBC] count, absolute neutrophil count [ANC], absolute lymphocyte count [ALC], mean corpuscular volume [MCV], and platelet counts [PLT]) were also assessed. Analyses were performed between and within the four patient groups. Response was assessed using the 2006 International Working Group (IWG) criteria [34], as redefined for erythropoietic response by the IWG 2018 criteria [35]. Factors associated with the achievement of a favorable response and with OS were also investigated.

### 2.1. Statistical Analysis

Statistical analysis was performed using IBM SPSS v.24 software (SPSS Inc., Chicago, IL, USA). Median and range were used to describe patient population characteristics. For variable comparisons between groups, the one-way analysis of variance (ANOVA) or Kruskal–Wallis H tests were employed, depending on the distribution, followed by the appropriate post-hoc tests (Bonferroni or Mann–Whitney U test). Wilcoxon signed-rank tests were used to compare parameters before and after treatment. Associations between categorical variables were analyzed using the chi-square (χ^2^) test. Linear regression models were constructed to evaluate the effect of baseline parameters (age, gender, time interval from initial diagnosis to lenalidomide treatment start, Hb, MCV, WBC, ANC, ALC, platelets, BM blast cell percentage, BM fibrosis, serum ferritin, and RBC transfusion dependence) on the type of response and on Hb increase. Coefficients with *p* < 0.200 were included in the models. Multivariate multinomial logistic regression models were constructed to examine associations between: (1) pre-treatment transfusion dependency or (2) response to treatment within a group of patients and response to treatment and baseline features. Survival analysis was performed by constructing Kaplan–Meier survival curves. Differences between curves were evaluated using the log-rank (Mantel–Cox) test. Moreover, Cox regression models were built to assess the association between survival time and baseline features. The statistical significance level was set at *α* = 0.05.

### 2.2. Ethics Approval and Consent to Participate

This study was conducted in accordance with the ethical principles of the Declaration of Helsinki that are consistent with Good Pharmacoepidemiology Practices and the applicable laws and regulations of Greece. As this was a retrospective analysis that used de-identified (anonymized) data, patients were not required to provide formal Consent to Release Information forms for the current analyses, beyond the original consent provided by those who agreed to be included in the EAKMYS.

## 3. Results

### 3.1. Analysis on an Intention to Treat Basis

According to the WHO 2016 classification, 153 patients (64.3%) had del(5)q syndrome, 34 (14.3%) had Refractory Anemia with Excess of Blasts (RAEB) type-1, 21 (8.8%) had RAEB type-2, 20 (8.4%) had Refractory Anemia with Multilineage Dysplasia, with or without ringed sideroblasts, 5 (2.1%) had refractory cytopenia with unilineage dysplasia, 3 (1.3%) had AML, and the remaining 2 (0.8%) had Chronic Myelomonocytic Leukemia (CMML). According to the most recent WHO-2022 classification, among the 238 patients who started lenalidomide treatment, 173 (72.7%) had MDS-del(5)q, 55 (23.1%) had an excess of marrow blasts (MDS-IB1 N = 34~14.3%, MDS-IB2 N = 21~8.4%), 7 (2.9%) had other types of MDS without an excess of marrow blasts (MDS-LB, multilineage dysplasia N = 6, Fibrotic MDS N = 1), and 3 patients (1.2%) had acute myelogenous leukemia. Baseline patient characteristics, disease classification, and prognostic categorization according to IPSS, WPSS, and IPSS-R are presented in Table 1.

### 3.2. Analysis on Evaluable Patients

Among 218 evaluable patients analyzed, 146 (67.0%) had 5q− syndrome and 72 (33.0%) had other MDS types associated with del(5)q, of whom 31 (40.8%) had isolated del(5)q. Thus, a total of 177 patients (81.2%) exhibited del(5)q as the only cytogenetic abnormality (Groups I-II), 20 (9.2%) had one additional cytogenetic abnormality over del(5)q (Group-III), and the remaining 21 (9.6%) exhibited >1 additional abnormality (Group-IV). There were no significant inter-group differences with regard to demographics and time from diagnosis to treatment start, with the exceptions of a higher female to male ratio in Group-I and a lower age at diagnosis in Group-II (Table 1). According to the newer WHO (2022) classification, 163 patients (74.8%, all from Group-I, 2 from Group-II, and 15 from Group-III) were classified as MDS-del(5)q, 5 patients as MDS-LB non-del(5)q (2.3%), 28 patients (12.8%) as MDS-IB1, 20 patients (9.2%) as MDS-IB2, and 2 patients (0.9%) as AML.

The majority of patients in Group-I had low/very low risk disease, according to IPSS, WPSS, or Revised IPSS (IPSS-R) scores. Group-III exhibited a slightly higher risk profile, with most patients classified as low to intermediate risk. Group-II and Group-IV had higher risk profiles, with patients in Group-IV exhibiting the highest risk (Table 1). The three patients with AML were elderly and unfit or unsuitable for systemic chemotherapy, and in view of the presence of del(5)q in their karyotype, it was considered by the treating physicians that they should be initially exposed to lenalidomide before proceeding to other treatment modalities, such as hypomethylating agents. Lenalidomide has previously been used in such instances, sometimes inducing impressive remissions [36,37]. 

No significant difference in the pretreatment duration of transfusion dependence between the four patient groups was noted (Kruskal–Wallis H test, *p* = 0.971). Lenalidomide dose was adapted in 84/218 evaluable patients (38.5%) for reasons of toxicity or intolerance. There is already strong evidence that this drug exerts excellent activity, even when it is used in lower doses than those recommended in the initial approval studies, or in alternative day regimens [38,39]. 

The most common cytogenetic abnormalities, on top of del(5)q, identified in patients of Group-III and Group-IV were trisomy 8 (n = 14 or 28%, in 6 as the one and only additional abnormality and in the remaining 7 accompanied by other abnormalities), del(20)q (n = 5 or 10%, in 4 as the one and only additional abnormality), monosomy 7 or del(7)q (n = 6 or 12%, in all in association with additional abnormalities), monosomy 18 (n = 4 or 8%, in all cases as part of complex additional abnormalities), trisomy 21 (n = 3 or 6%, in 2 as part of additional abnormalities), and del(13)q14 (n = 3; Appendix A).

### 3.3. Hematological Response

A total of 218 (91.6%) patients were evaluable for response. Twenty patients were not evaluable for response since they did not complete at least two cycles of treatment due to early death (n = 6, of which, sudden cardiac death [n = 2] and severe fatal infection [n = 4]), grade 4 myelosuppression with prolonged hospitalization and patient refusal to continue treatment (n = 5), severe (grade 4) skin toxicity (n = 4), disease progression, and severe clinical deterioration (n = 3), not otherwise specified (n = 2). Among evaluable patients, a major hematological response (MR) was achieved by 146 (67.9%), and particularly by 114/146 (78.1%) from Group-I, 18/31 (58.1%) from Group-II, 10/20 (50.0%) from Group-III, and by 4/21 (19.0%) from Group-IV (Table 2). Corresponding overall response rates (ORRs) were 81.2% in the overall evaluable patient population and 88.4%, 80.7%, 80.0%, and 33.3%, respectively, in each of the four patient groups (Table 2). Collectively, a favorable response was obtained by 48/72 (66.7%) patients in Groups II-IV, and, among them, by 31/50 (65.9%) evaluable patients with an excess of BM blasts (n = 29) or AML (n = 2). Appendix A shows the proportions of patients who exhibited normalization of their hematological parameters following lenalidomide treatment, according to the type of response achieved, in each patient group and overall.

The median time to achieve an MR (4.2 months [95% confidence interval (CI): 3.9–4.6]) was significantly longer than the median time to achieve a Minor response (MiR) (3.6 months [95% CI: 1.2–7.5], Mann–Whitney U test, *p* = 0.012). The median time to any response was 4.0 months (95% CI: 1.2–10.7). No significant difference in time to achieve a favorable response between the four patient groups was observed (Mantel–Cox test, *p* > 0.050).

Overall, the median duration of any response was 31.1 months (95% CI: 24.8–38.2), 34.0 in Group-I, 17.1 in Group-II, 45.6 in Group-III, and 22.0 months in Group-IV (no significant difference between groups, Mantel–Cox test, *p* > 0.050; Appendix A). The median duration of response for patients who achieved MR or MiR/hematological improvement was 34.7 months (95% CI: 29.2–40.2) and 16.4 months (95% CI: 0.4–32.4), respectively (Mantel–Cox, *p* < 0.001). Median duration of response was lower for patients with an excess of BM blasts than for those without (18.4 vs. 34.7 months, *p* = 0.022). However, when the comparison of the duration of response was restricted to patients who achieved an MR, no significant intergroup difference was observed even among patients with an excess of BM blasts or additional cytogenetic abnormalities (Mantel–Cox test, *p* > 0.050).

### 3.4. Predictors of Hematological Response

Under multivariate logistic regression, analyzing response to treatment according to patient group, the probability of achieving MR was significantly lower in Group-IV than in Group-I (odds ratio [OR] 0.04, 95% CI: 0.01–0.15, *p* < 0.001, Table 3). Collectively, patients not classified as 5q− syndrome (Groups II + III + IV) had a lower probability of achieving MR than those with the 5q− syndrome (Group-I, OR = 0.20, 95% CI: 0.10–0.42, *p* < 0.001). However, this was largely attributable to patients in Group-IV, who had a significantly lower probability of achieving MR than patients with isolated del(5)q or del(5)q plus only one additional abnormality, not affecting chromosome 7 (Groups II + III) (OR = 0.10, 95% CI: 0.03–0.38, *p* = 0.001, Table 3). On an ITT basis, patients in Group-I had a higher probability of being either completely or partially transfusion-independent at baseline than patients in Group-II or Group-IV, or those in Groups-II + III + IV combined; additionally, transfusion-independent patients tended to have a higher probability of achieving MR (OR = 1.94, 95% CI: 0.71–4.86). However, the duration of pretreatment complete transfusion dependence was not different across the four groups (Kruskal–Wallis *p* = 0.971) and did not affect the achievement of response as the logistic regression model was not significant (*p* = 0.921). Baseline parameters, which were strongly associated with the achievement of response irrespective of patient group, included a higher MCV (higher probability of achieving MR [OR = 1.05, 95% CI: 1.02–1.09, *p* = 0.002]) and a higher percentage of BM blasts (lower probability of achieving MR [OR = 0.86, 95% CI: 0.79–0.93, *p* < 0.001]; Appendix A).

### 3.5. Changes in Hematological Parameters

Macrocytic MCV (>95 fL) was found in 73.5% of the patients overall, and in 82.4%, 79.4%, 80.8%, and 44.0% in Groups-I, -II, -III, and -IV, respectively. Corresponding percentages for high-normal or frankly increased (>300 × 10^9^/L) platelet counts were 29.8%, 35.9%, 23.5%, 24.0%, and 8.0%, respectively. On the other hand, thrombocytopenia (<100 × 10^9^/L) was encountered in 13.9% of the patients overall and in 5.2%, 17.6%, 19.2%, and 47.6% in Groups-I, -II, -III, and -IV, respectively. Thus, baseline MCV and platelet count were significantly lower in Group-IV than in the other groups (Kruskal–Wallis test, *p* = 0.003 and *p* < 0.001, respectively), whereas Hb, WBC, ANC, and ALC were comparable in all groups. Lenalidomide had a variable impact on the hematological parameters of patients who responded favorably. Whereas Hb increased in responding patients in all groups to a similar extent (Kruskal–Wallis test, *p* = 0.254), the majority of responding patients exhibited a >5 fl decrease in their MCV (146/177 [~82.5%]). MCV fell ≥10% from baseline in 98 patients (55.4%) and returned to the normal range (<95 fL) in 97 patients (54.8%). Post-treatment platelet counts were significantly reduced across the whole patient population, as well as in Groups-I, -III, and -IV (Wilcoxon test, *p* < 0.001). To investigate this finding further, we split the entire evaluable patient group (218 patients) into three subgroups, according to baseline platelet count Group A: thrombocytopenic (platelets < 100 × 10^9^/L, N = 24); Group B: normal platelets (platelets 100–400 × 10^9^/L, N = 161); and Group C: thrombocytotic (platelets >400 × 10^9^/L, N = 33) and compared again all hematological parameters at baseline and at best response. Whereas an Hb increase and MCV correction were observed in all three platelet-defined subgroups, as previously described, post-treatment platelet counts were significantly increased only in Group A (*p* = 0.001), but were decreased in Groups B and C (both *p* < 0.001).

To more precisely investigate the effect of lenalidomide on the dysplastic clone, we then restricted the comparison of the various hematological parameters in the responded patient population, which consisted of 177 subjects. Thus, in the whole group of these subjects, a favorable response to lenalidomide treatment was associated with a significant increase in the mean hemoglobin value (*p* < 0.001), total WBC count (*p* = 0.030), and absolute lymphocyte count (*p* = 0.011), and with a significant drop in MCV (*p* < 0.001), platelet count (*p* < 0.001), bone marrow blast cell percentage (*p* < 0.001), and serum ferritin levels (*p* < 0.001). Similar significant differences for Hb, MCV, platelet count, BM blasts, and serum ferritin levels were also observed in patients of Group-I, the last difference indicating a shift towards more effective erythropoiesis. An interesting finding was the very significant drop of mean bone marrow blast cell percentage in the whole group of responded patients and in Group-I, but particularly in Group-II, from a pretreatment value of 10.5 ± 6.3% to 2.7 ± 2.5% post-treatment, implying that among responded patients, the course towards AML can be modified or at least delayed by lenalidomide. The drop in BM blasts for patients of Group-III and Group-IV, exhibiting additional cytogenetic abnormalities, although prominent, did not reach the level of statistical significance, most probably due to the low number of patients. In Group-IV, consisting of only 7 patients, the decrease in platelet count was not statistically significant, and the only significant post-treatment change was the increase in Hb levels. Table 4 shows the observed post-treatment changes in the hematological parameters in the whole responded patient population, as well as in each of the four groups separately.

In linear regression models, Hb increase was positively associated with baseline MCV values, the duration of response, complete transfusion dependence at baseline, and belonging to Group-I, and negatively with baseline Hb and gender (females exhibited a smaller Hb increase). MCV decrease was positively associated with baseline Hb and complete transfusion dependence and negatively with baseline MCV and ALC values, belonging to Group-I or Group-II, duration of response, duration of complete transfusion dependence, and age at diagnosis.

### 3.6. Transfusion Independence

At baseline, 95 out of the 218 evaluable patients (43.6%) were completely transfusion-dependent (requiring more than 2 RBC units per month to stabilize their condition), 68 patients (31.2%) were partially transfusion-dependent (requiring no more than 2 units per month), and the remaining 55 patients (25.2%) were not transfusion-dependent. Complete transfusion dependence was more frequent in patients of Group-IV (68%) and Group-II (50%). Among responders, 103 of the 116 regularly transfused patients at baseline (88.8%) achieved transfusion independence following lenalidomide treatment, and the remaining 13 patients (11.2%) decreased their transfusion requirements. A comparison of the frequency of patients achieving transfusion independence across the four patient groups showed no difference in their distribution (Fisher’s exact test, *p* = 0.824). This finding indicates that if a patient responds favorably to lenalidomide, the likelihood of achieving complete transfusion independence is not influenced by the presence of one or more additional cytogenetic abnormalities over the del(5)q clone, or in other words, that lenalidomide can restore effective erythropoiesis, even in patients with additional cytogenetic abnormalities, over del(5)q. However, as previously mentioned, baseline transfusion dependence did not influence the outcome of treatment, although in Group-I only, pretreatment transfusion independence was associated with a higher probability of achieving an MR. Overall, 56/60 patients from all Groups with complete transfusion dependence (93.3%) and 47/56 with partial transfusion dependence (83.9%) became transfusion-independent. Results of the evolution of transfusion requirements, in relation to treatment and to patient group, are shown in Table 5.

### 3.7. Cytogenetic Response

A cytogenetic re-evaluation was performed in 114/218 (52.3%) patients at a median of 7.2 months (range 4.5–15.2) post-treatment; meaningful results were available for 108 patients (49.5%). A complete cytogenetic response was achieved by 41 patients (38.0%) overall, 33/78 (42.3%) in Group-I, 5/14 (35.7%) in Group-II, 1/7 (14.3%) in Group-III, and 2/9 (22.2%) in Group-IV (chi-square test for difference between groups, *p*: not significant). A partial cytogenetic response was observed in 46 patients (42.6%) overall, and particularly in 35/78 (44.9%) from Group-I, 6/14 (42.9%) from Group-II, 3/7 (42.9%) from Group-III, and in 2/9 (22.2%) from Group-IV; the remaining 21 patients (19.4%) showed no cytogenetic response. Thus, overall, any cytogenetic response was observed in 87 patients (80.6%), and particularly in 68/78 (87.2%) from Group-I, in 11/14 (78.6%) from Group-II, in 4/7 (57.1%) from Group-III, and in 4/9 (44.4%) from Group-IV (chi-square test for difference between groups, *p* = 0.025). There was a positive correlation between the achievement of a hematological and a cytogenetic response (Kendall’s tau = 0.371, *p* < 0.001), and all the patients who achieved a complete cytogenetic response and 40/46 (87%) of those who achieved a partial cytogenetic response exhibited a major hematological response; however, nine patients who achieved a major hematological response showed no cytogenetic response. Among evaluable patients, who were not thrombocytopenic at baseline (platelet counts > 200 × 10^9^/L, n = 101), the achievement of complete cytogenetic response was associated with a greater post-treatment reduction in platelet count than the achievement of partial or no response (41 ± 23% vs. 19 ± 40%, *p* = 0.031).

### 3.8. Survival

Median OS in the whole patient population was 70.2 months (95% CI: 54.2–86.2). OS was significantly longer in Group-I (median 86.1 months [95% CI: 70.0–105.2]) than in Group-II (median 51.8 months [95% CI: 28.0–75.6], *p* = 0.046), Group-III (median 73.3 months [95% CI: 43.0–103.6], *p* = 0.052), or Group-IV (median 20.3 months [95% CI: 13.3–27.3], *p* < 0.001, Figure 1A). Estimates for OS did not differ between Group-II and Group-III; when these groups were combined, median OS (52.3 months [95% CI: 22.6–82.0]) was significantly longer than that for Group-IV (*p* < 0.001), but lower than that for Group-I (*p* = 0.013, Figure 1B).

Median OS in evaluable patients was estimated at 76.6 months (95% CI: 62.4–90.8) (Figure 2A), while the corresponding OS among evaluable patients with 5q− syndrome was 88.7 months (95% CI: 68.0–109.4) (Figure 2B). Patients who achieved MR exhibited significantly prolonged OS (median 95.4 months [95% CI: 79.2–111.6]) vs. those who achieved MiR (median 38.0 months [95% CI: 23.3–52.7]) or no response (median 42.0 months [95% CI: 20.9–64.1], *p* < 0.001), both overall and among patients with 5q− syndrome in particular (median 103.0 months [95% CI: 82.8–123.2], *p* = 0.009 and *p* = 0.027 for MiR and no response, respectively). Under Cox regression hazard (multivariate) analysis, factors associated with shorter OS were older age at treatment start, higher WBC counts (borderline significance), higher BM blasts, worse WHO-defined performance status, achievement of MiR or no response, shorter interval between initial diagnosis and lenalidomide treatment start, and complete transfusion dependency at baseline (Table 6).

### 3.9. Safety

We did not observe any unexpected or frequent adverse events in the whole patient group or in the four subgroups with specific cytogenetic features. Skin reactions/toxicity were observed in 67 cases (28.2%) but were of grade 4 in 11, of whom 10 required hospitalization. Five of the 20 non-evaluable patients discontinued lenalidomide due to early and unacceptable skin toxicity. Thromboembolic events were encountered in 19 patients and were considered minor in 9 and major in 10 patients, including acute myocardial infarction in 4 cases, pulmonary embolism in 3, and cerebrovascular attacks in 3. Sudden cardiac death soon after treatment onset was observed in 2 patients and occurred at a later stage in an additional 3 patients. Two hundred and seven patients experienced at least one infectious episode during treatment, which required hospitalization in 67 patients. Neutropenia was observed in 56 patients, mainly and more profoundly in the Group-III and Group-IV, whereas symptomatic thrombocytopenia (<30 × 10^9^/L) was noticed in 27 patients, mainly in those who were already thrombocytopenic at baseline. Severe hemorrhagic manifestations requiring platelet transfusions were observed in 6 patients, particularly in those with higher-risk disease. Other types of toxicity included neuropsychiatric manifestations/disease deterioration in 9 patients, renal impairment in 14, which was reversible in the majority of cases, hepatotoxicity mainly of minor importance in 11 patients, various rheumatological manifestations in 27 patients, and other autoimmune manifestations, excluding skin and rheumatological manifestations in 6.

Disease progression was observed in 81 patients, in the form of a more aggressive type of MDS in 40 and as a direct leukemic transformation in the remaining 41 patients.

## 4. Discussion

This retrospective analysis of 238 patients with del(5)q, treated with lenalidomide and identified in the Greek National MDS database (EAKMYS), is one of the largest studies, providing real-world data, and one of the very few, reporting separate results on patients with del(5)q plus additional cytogenetic abnormalities. The results obtained suggest that lenalidomide might be an effective treatment option, not only for patients with the 5q− syndrome, but also for MDS patients with isolated del(5)q, as well as for those with del(5)q plus one additional cytogenetic abnormality, not affecting chromosome 7. Of note, high response rates to lenalidomide were seen even among patients with an excess of BM blasts or, in a few instances, frank AML, which represent patient groups that, until now, have not been extensively analyzed in similar efficacy reports. Sporadic cases of high-risk MDS with the del(5)q abnormality, or of patients with del(5)q plus additional cytogenetic abnormalities, exhibiting favorable responses to lenalidomide do exist in some smaller retrospective studies [40,41,42], but have not been carefully grouped and systematically analyzed. Such patients are, for example, 5 and 6, respectively, out of the 50 presented in the Austrian study [43] and 7 and 9, respectively, out of 55 analyzed in the Czechish study [44], 12 out of 99 in a collaborative, mainly Italian study [45], and 13 out of 77 patients in the Chinese study [46]. In the initial report of the GFM, among 95 patients analyzed, only 36 had 5q- syndrome, whereas 24 patients with RAEB-1 were treated, of whom 3 attained a complete cytogenetic response, but no other information is provided for this group of patients [47]. In another analysis of the same patient group, it has been reported that lenalidomide does not increase the risk of leukemic transformation, but again no information on the response rate and duration of response is provided [48]. 

Both major and overall response rates reported in our study compare favorably with those previously reported in the approval study MDS-003 [49], as well as with the rates reported by another real-world registry study of patients with del(5)q, with or without additional chromosomal abnormalities, who received ≥1 cycle of lenalidomide treatment [5]. Results are also in line with those of clinical trials investigating the efficacy of lenalidomide in patients with IPSS low/intermediate-1 risk del(5)q MDS [11,50], as well as with a Chinese study, in which the presence of additional cytogenetic abnormalities did not influence response [46]. Response rates of patients from group-I and group-II are pretty similar to those reported in the Polish study in their uniform group of 36 patients [50]. However, in the current study, the median follow-up of responded patients is very extended (39 months, range 6.6–162 months), which is even longer than the updated results of the MDS-003 study (3.1 years) published in 2014 [51]. Moreover, in our study, lenalidomide has demonstrated efficacy, not only in patients with 5q− syndrome but also induced ORRs of >75% in evaluable patients with other types of MDS, exhibiting isolated del(5)q or del(5)q plus one additional cytogenetic abnormality, including higher-risk patients. Although the number of patients treated in these groups was not very high (31 and 26, respectively), to our knowledge this is one of the larger retrospective studies, analyzing results of lenalidomide treatment in patient groups, defined according to both BM blast percentage and complexity of additional cytogenetic abnormalities, on top of del(5)q. Several case reports have described the efficacy of lenalidomide in occasional patients with del(5)q, plus one additional abnormality [24,25,26,27]. In the Chinese study, all 13 patients without del(5)q alone exhibited only one additional cytogenetic abnormality [46]. Thus, messages emerged from such previous small reports and results obtained from the current analysis confirming the previous suggestion of Zeidan et al., that such patients should be initially exposed to lenalidomide rather than to hypomethylating agents (azacitidine or decitabine), which should be reserved for second-line treatment [52,53]. 

The finding that the presence of >1 additional cytogenetic abnormality negatively impacts response to lenalidomide in patients with del(5)q is in line with the findings of other real-world studies [5,6] and with a clinical study investigating lenalidomide as monotherapy in higher-risk MDS patients with complex del(5)q karyotypes [54]. 

A complete cytogenetic response was seen in 38% of our patients, who underwent re-evaluation, which is somewhat higher than what is reported in the previously mentioned registry study [5]. The authors of that study attributed their findings to heterogeneity of treatment schedules, methods of lenalidomide administration, and patient dosage adjustments. In our study, all patients initially received the recommended lenalidomide dose of 10 mg for 21 consecutive days every 4 weeks; dosage adjustments were required in fewer than 40% of the patients, and in all cases, this was attributed to intolerance of the initial lenalidomide dosage. A similar percentage of dose adaptation has also been described in other studies [38,39], but dose reduction following toxicity appears not to influence treatment outcome [55]. However, Mikkael Sekeres et al., performing a meta-analysis of the MDS-003 and MDS-004 studies on 217 patients, found that starting treatment with a lenalidomide dose lower than 10 mg × 21 days, q28 days, or the unjustified dose reduction in the absence of severe myelotoxicity might be associated with a worse outcome [56]. Another potential explanation for the higher complete cytogenetic response rate detected in our study is that several patients underwent a BM re-evaluation at a relatively later time point even after 12 of 15 months of treatment, and hence they had been exposed to treatment for a prolonged time period.

Lenalidomide acts by inhibiting the expression of key genes involved in the pathophysiology of the del(5)q clone, concurrently enhancing marrow re-population with non-clonal cells [19,41,49,57,58]. In patients with isolated del(5)q, the maternal clone may generate subclones bearing additional numerical or qualitative cytogenetic abnormalities that are associated with increasing disease aggressiveness [2]. Our findings imply that, in many of such cases, the dominant pathophysiology might be governed by the presence of the maternal clone, which can be effectively inhibited by lenalidomide [59]. Thus, even when treatment starts at a later disease stage, when an excess of BM blasts has been developed, remission can be achieved. In support of the above-mentioned speculation is the comparable time needed to achieve a favorable response in patients of any cytogenetic group, either exhibiting del(5)q alone or del(5)q plus one or more cytogenetic abnormalities.

Two common hematological findings in MDS patients with del(5)q are macrocytic anemia and mild-to-moderate thrombocytosis. These findings appear to be strongly associated with the presence of the neoplastic clone, and, in the current study, the eradication or the substantial shrinkage of the clone resulted in MCV correction and in a significant decrease in platelet counts to the normal range or transiently to mild thrombocytopenic levels (100–150 × 10^9^/L). This decrease in platelet count post-lenalidomide treatment could be mainly attributed to the correction of the mild-to-moderate thrombocytosis that is commonly associated with del(5)q and should not be confused with a disease-related mild thrombocytopenia, present before lenalidomide treatment start, which has been recognized as an unfavorable factor for response to treatment. Thus, it appears that both macrocytic MCV and high-normal or mildly-moderately increased platelet counts might reflect dyserythropoiesis and dysmegakaryopoiesis, induced by the del(5)q clone, and, therefore, could represent potential markers of low-risk disease activity. The correction of these markers during lenalidomide therapy might indicate early evidence of response and could represent reliable simple markers of clonal elimination, overcoming the need for the application of more complex and sophisticated molecular signatures [60,61]. In support of the prognostic significance of the baseline platelet count are also the results of the Spanish study on 52 patients [6]. Other, well-recognized adverse prognostic factors for the achievement of favorable response among patients with del(5)q are age, performance status, transfusion-dependency, thrombocytopenia, increased BM blasts, and complex cytogenetics at baseline [62,63], as well as the presence of TP53 mutations [64]. However, in the Mayo Clinic analysis, transfusion need at baseline turned out to be a favorable prognostic factor for response, whereas previous treatment with ESAs had a negative impact on response to lenalidomide [65]. In the Italian retrospective analysis, additional prognostic factors independently predicting an unfavorable outcome were also higher neutrophil count and elevated serum LDH levels, whereas achievement of transfusion independence following treatment was associated with a better outcome [45]. 

As expected, the median duration of response was significantly higher in patients achieving an IWG-defined major response than in those achieving minor response/hematological improvement, and the duration of overall response was consistently longer in patients with 5q− syndrome than in other patient groups (Appendix A). However, the absence of a significant intergroup difference in MR duration implies that, in patients in whom lenalidomide is able to effectively reverse the development of the neoplastic clone, the duration of response might be comparable, irrespective of whether an excess of BM blasts or additional cytogenetic abnormalities were present at baseline.

Finally, of importance is the finding in this study that a shorter interval between initial diagnosis and start of lenalidomide treatment was independently associated with longer OS. This finding, which has also been reported in several previous studies, raises the question of whether the recommendation that only symptomatic or transfusion-dependent patients with MDS, exhibiting del(5)q, should be treated with lenalidomide, following ESA failure, is correct. Our finding favors an earlier treatment start, with the aim of more effectively shrinking the neoplastic clone, thus limiting its ability to expand and/or become resistant to lenalidomide, as this has been delineated also by other retrospective studies [20]. 

Regarding safety, although initially some concerns about a potential enhancement of disease progression had been raised, it soon became clear that this is not true and that such a progression represents part of the natural course of the disease and is not related to lenalidomide treatment [66,67]. 

### Strengths and Limitations of This Study

Although this was a retrospective study of registry data, and for some of the defined groups the number of patients is not high enough to draw safe conclusions, the data obtained from the EAKMYS database provides a detailed description of a large patient group analyzed, representative of the Greek MDS patient population as a whole, in which diagnosis was confirmed by marrow morphology/trephine biopsy. This study analyzes one of the largest uniform groups of del(5)q patients reported in the literature by now, and evaluable patients had a very extended median follow-up of 39 months. Notable numbers of patients with del(5q) other than 5q− syndrome, treated with lenalidomide, were also included and have allowed us to identify useful and easily obtained predictors of response, not previously described. Follow-up was very extended and as complete and thorough as possible, with very few patients being lost from follow-up. Data were collected and analyzed on an ITT basis in patients who received at least one dose of lenalidomide; response was evaluated among patients who completed at least two cycles of therapy.

## 5. Conclusions

Lenalidomide was highly effective, not only in patients with 5q− syndrome but also in those with isolated del(5)q or with del(5)q plus one additional cytogenetic abnormality despite the small number of patients in this group. High response rates were seen even in patients with an excess of blasts. Of particular interest is the finding that the duration of MR following lenalidomide was comparable between all patient groups, implying that lenalidomide could represent a potential initial treatment option for these patient groups, prior to the use of hypomethylating agents.

## Figures and Tables

**Figure 1 cancers-17-01388-f001:**
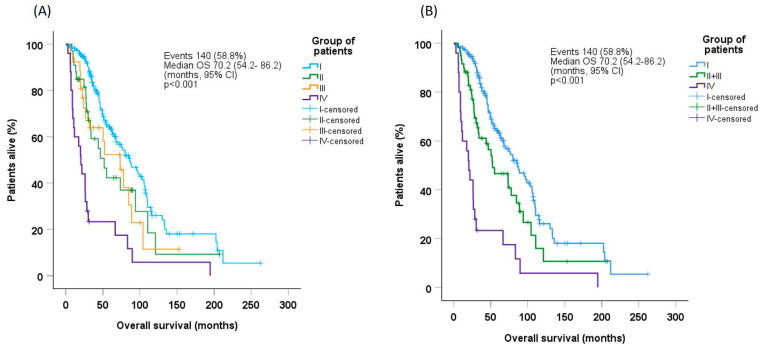
Overall survival probability in (**A**) the whole patient population and (**B**) Groups I, combined (II + III), and IV, according to patient group. CI, confidence interval; OS, overall survival.

**Figure 2 cancers-17-01388-f002:**
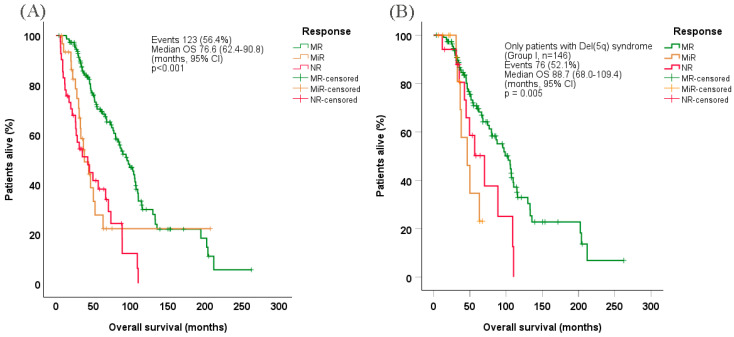
Overall survival probability in (**A**) all evaluable patients and (**B**) Only patients with Del(5q) syndrome (Group I), according to response type. CI, confidence interval; Del(5q), deletion of chromosome 5q; MR, major response; MiR: minor response; NR, no response; OS, overall survival.

**Table 1 cancers-17-01388-t001:** Baseline patient characteristics overall and according to group.

	Group-I	Group-II	Group-III	Group-IV	Total
WHO 2016 Classification/ Cytogenetics	5q− Syndrome	Other MDS with Isolated Del(5)q	Del(5)q + 1 Additional Abnormality	Del(5)q + >1 Additional Abnormality	All Patients
N =	153	34	26	25	238
Female/male, N	121/32	21/13	18/8	15/10	175/63
Female/male ratio	3.78	1.58	2.25	1.50	2.78
Median age at diagnosis, years (range)	74 (49–95)	71 (48–85)	76 (59–88)	75 (55–82)	74 (48–95)
Median age at treatment start, years (range)	76 (49–96)	71 (50–86)	76 (59–88)	77 (55–83)	75 (49–96)
Median time from diagnosis to treatment start, months (range)	8.6 (0.7–184.4)	10.8 (0.8–119.9)	8.3 (0.2–61.6)	4.1 (1.4–110.6)	8.1 (0.2–184.4)
**WHO 2016 classification, n (%)**					
Del(5)q syndrome	153 (100.0)	0	0	0	153 (64.3)
RA	0	0	5 (19.2)	0	5 (2.1)
RCMD ± RS	0	1 (2.9)	13 (50.0)	6 (24.0)	20 (8.4)
RAEB1	0	22 (64.7)	6 (23.1)	6 (24.0)	34 (14.3)
RAEB2	0	8 (23.6)	2 (7.7)	11 (44.0)	21 (8.8)
AML	0	2 (5.9)	0	1 (4.0)	3 (1.3)
CMML	0	1 (2.9)	0	1 (4.0)	2 (0.8)
**WHO 2022 classification, n (%)**					
MDS-Del(5)q	153 (100.0)	2 (4.9)	18 (69.2)	0	173 (72.7)
MDS-LB	0	0 (17.1)	0	7 (28.0)	7 (2.9)
MDS-IB1	0	22 (53.7)	6 (23.1)	6 (24.0)	34 (14.3)
MDS-IB2	0	9 (22.0)	2 (7.7)	10 (40.0)	21 (8.8)
AML	0	1 (2.4)	0	2 (8.0)	3 (1.3)
**IPSS risk group, n (%)**					
Low	128 (83.7)	0	0	0	128 (53.8)
Intermediate-1	25 (16.3)	25 (73.5)	23 (88.5)	7 (28.0)	80 (33.6)
Intermediate-2	0	7 (20.6)	1 (3.8)	6 (24.0)	14 (5.9)
High	0	2 (5.9)	2 (7.7)	12 (48.0)	16 (6.7)
**WPSS risk group, n (%)**					
Very low	71 (46.5)	0	1 (3.8)	0	72 (30.3)
Low	72 (47.0)	2 (5.9)	6 (23.1)	1 (4.0)	81 (34.0)
Intermediate	10 (6.5)	10 (29.4)	10 (38.5)	3 (12.0)	33 (13.9)
High	0	20 (58.8)	8 (30.8)	8 (32.0)	36 (15.1)
Very high	0	2 (5.9)	1 (3.8)	13 (52.0)	16 (6.7)
**IPSS-R risk group, n (%)**					
Very low	23 (15.0)	0	1 (3.8)	0	24 (10.1)
Low	111 (72.6)	3 (8.8)	15 (57.7)	0	129 (54.2)
Intermediate	19 (12.4)	19 (55.9)	5 (19.2)	1 (4.0)	44 (18.5)
High	0	10 (29.4)	5 (19.2)	9 (36.0)	24 (10.1)
Very high	0	2 (5.9)	0	15 (60.0)	17 (7.1)

Abbreviations: AML, acute myeloid leukemia; CMML, chronic myelomonocytic leukemia; Del(5q), deletion of chromosome 5q; IPSS, International Prognostic Scoring System; IPSS-R, Revised IPSS; MDS, myelodysplastic syndromes; RA, refractory anemia; RAEB, refractory anemia with excess blasts; RCMD, refractory cytopenia with multilineage dysplasia; RS, ringed sideroblasts; WHO, World Health Organization; WPSS, WHO classification-based Prognostic Scoring System.

**Table 2 cancers-17-01388-t002:** Response to lenalidomide treatment overall and according to patient group.

		No Response			Minor Response			Major Response			Overall Response		
Group (ITT Population)	Evaluable Patients ^a^	N	% of Eval.	% ITT	N	% of Eval.	% ITT	N	% of Eval.	% ITT	N	% of Eval.	% ITT
I (n = 153)	146	17	11.6	11.1	15	10.3	9.8	114	78.1	74.5	129	88.4	84.3
II (n = 34)	31	6	19.4	17.6	7	22.6	20.6	18	58.1	50.0	25	80.6	73.5
III (n = 26)	20	4	20.0	15.4	6	30.0	23.1	10	50.0	38.5	16	80.0	61.5
IV (n = 25)	21	14	66.7	56.0	3	14.3	12.0	4	19.0	16.0	7	33.3	28.0
Total (n = 238)	218	41	18.8	17.2	31	14.2	13.0	146	67.0	61.3	177	81.2	74.4

Abbreviation: AE, adverse event; ITT, intention to treat. **^a^** 20 patients (Group-I: [n = 7], Group-II: [n = 3], Group-III: [n = 6], and Group-IV: [n = 4]) were considered not evaluable for response after treatment with lenalidomide because they did not complete at least two cycles of treatment as a result of severe toxicity/intolerance, disease progression, or death, unless they responded earlier (3 patients). Reasons for early treatment discontinuation were grade 4 myelotoxicity with infection (n = 8); severe cardiac AE (n = 5); early death (during febrile neutropenia [n = 3] or sudden cardiac death [n = 2]); grade 4 skin toxicity (n = 4); severe clinical deterioration or treatment intolerance, not otherwise specified (n = 4); acute renal failure (n = 2, in one patient with an infection/sepsis); and rapid disease progression following initial myelosuppression (n = 2). More than one reason was reported in 11 patients.

**Table 3 cancers-17-01388-t003:** Factors related to response to treatment in evaluable patients (multivariate logistic regression models).

	None (n)	Minor (n)	Major (n)	OR	95% CI	*p* Value
**Group**						
Group-I	17		114	1.00		
Group-II	6		18	0.45	0.16–1.29	0.135
Group-III	4		10	0.37	0.11–1.32	0.127
Group-IV	14		4	0.04	0.01–0.15	**<0.001**
Group-I	17	15		1.00		
Group-II	6	7		1.32	0.36–4.82	0.672
Group-III	4	6		1.70	0.40–7.20	0.471
Group-IV	14	3		0.24	0.06–1.01	**0.** **052**
**Patients with 5q− syndrome vs. all others**						
5q− syndrome (Group-I)	17		114	1.00		
Other MDS with Del(5q) (Groups II + III + IV)	24		32	0.20	0.10–0.42	**<** **0.001**
5q− syndrome (Group-I)	17	15		1.00		
Other MDS with Del(5q) (Groups II + III + IV)	24	16		0.76	0.30–1.93	0.559
**Patients with isolated Del(5q) or +1 abnormality vs. patients with >1 additional abnormality**						
MDS with isolated Del(5q)/+1 abnormality (Groups II + III)	10		28	1.00		
5q− syndrome (Group-I)	17		114	2.40	0.99–5.80	**0.0** **53**
Del(5q) plus > 1 abnormality (Group-IV)	14		4	0.10	0.03–0.38	**0.00** **1**
MDS with isolated Del(5q)/+1 abnormality (Groups II + III)	10	13		1.00		
5q− syndrome (Group-I)	17	15		0.68	0.23–1.99	0.481
Del(5q) plus >1 abnormality (Group-IV)	14	3		0.17	0.04–0.74	**0.** **018**

Abbreviations: CI, confidence interval; Del(5q), deletion of chromosome 5q; MDS, myelodysplastic syndromes; OR, odds ratio. Statistically significant differences are highlighted with bold figures.

**Table 4 cancers-17-01388-t004:** Changes in the hematological parameters among responders.

	Pre Len Tx	Post Len Tx	*p*
**All responded patients, N = 177**			
Hb (g/dL)	8.7 ± 1.2	12.7 ± 1.6	**<0.001**
MCV (fL)	103.2 ± 10.7	91.7 ± 9.3	**<0.001**
WBC (×10^9^/L)	4.41 ± 1.96	4.72 ± 2.26	**0.030**
ANC (×10^9^/L)	2.26 ± 1.41	2.34 ± 1.40	0.485
ALC (×10^9^/L)	1.58 ± 0.63	1.71 ± 0.61	**0.011**
PLT (×10^9^/L)	262 ± 142	166 ± 69	**<0.001**
Ferritin(ng/mL)	498 ± 611	375 ± 470	**<0.001**
BM blasts (%)	3.5 ± 3.6	1.6 ± 1.0	**<0.001**
**Group-I, N = 129**			
Hb (g/dL)	8.6 ± 1.2	12.8 ± 1.5	**<0.001**
MCV (fL)	103.3 ± 10.6	91.1 ± 9.3	**<0.001**
WBC (×10^9^/L)	4.48 ± 1.83	4.60 ± 1.58	0.456
ANC (×10^9^/L)	2.35 ± 1.45	2.29 ± 1.15	0.623
ALC (×10^9^/L)	1.58 ± 0.59	1.69 ± 0.59	0.056
PLT (×10^9^/L)	280 ± 149	172 ± 66	**<0.001**
Ferritin(ng/mL)	506 ± 668	336 ± 446	**<0.001**
BM blasts (%)	2.2 ± 1.2	1.5 ± 0.9	**<0.001**
**Group-II, N = 25**			
Hb (g/dL)	8.6 ± 1.2	12.4 ± 1.2	**<0.001**
MCV (fL)	103.2 ± 13.0	92.1 ± 10.5	**<0.001**
WBC (×10^9^/L)	3.81 ± 1.94	4.74 ± 2.37	**0.024**
ANC (×10^9^/L)	1.86 ± 1.33	2.41 ± 1.57	**0.038**
ALC (×10^9^/L)	1.42 ± 0.60	1.66 ± 0.55	**0.046**
PLT (×10^9^/L)	221 ± 104	170 ± 84	**0.013**
Ferritin(ng/mL)	531 ± 401	456 ± 520	0.489
BM blasts (%)	10.5 ± 6.3	2.7 ± 2.5	**<0.001**
**Group-III, N = 16**			
Hb (g/dL)	9.0 ± 0.8	12.4 ± 1.9	**<0.001**
MCV (fL)	104.6 ± 6.2	95.3 ± 7.4	**0.001**
WBC (×10^9^/L)	4.24 ± 1.45	4.61 ± 2.05	0.273
ANC (×10^9^/L)	2.01 ± 0.94	2.20 ± 1.38	0.433
ALC (×10^9^/L)	1.79 ± 0.69	1.88 ± 0.83	0.472
PLT (×10^9^/L)	221 ± 103	134 ± 64	**0.002**
Ferritin(ng/mL)	281 ± 263	241 ± 229	0.450
BM blasts (%)	3.3 ± 2.8	1.2 ± 1.0	0.086
**Group-IV, N = 7**			
Hb (g/dL)	8.6 ± 1.1	12.5 ± 2.5	**0.019**
MCV (fL)	99.2 ± 10.7	93.1 ± 8.7	0.053
WBC (×10^9^/L)	5.34 ± 3.81	7.09 ± 7.01	0.330
ANC (×10^9^/L)	2.51 ± 1.46	3.36 ± 3.39	0.474
ALC (×10^9^/L)	1.74 ± 1.05	1.79 ± 0.56	0.895
PLT (×10^9^/L)	175 ± 131	120 ± 46	0.280
Ferritin(ng/mL)	572 ± 478	1068 ± 548	0.148
BM blasts (%)	6.8 ± 4.9	2.3 ± 1.6	0.094

With bold figures only the statistically significant differences are highlighted.

**Table 5 cancers-17-01388-t005:** Evolution of RBC transfusion dependence with Lenalidomide treatment among responders.

Level of Transfusion Dependence	Pre-Len Treatment		Post-Len Treatment		*p*-Value *
**All patients**	** *N* **	**%**	** *N* **	**%**	
Complete	60	33.9	0	0.0	**<0.001**
Partial	56	31.6	13	7.3	
None	61	34.5	164	92.6	
**Group-I**					
Complete	42	32.6	0	0.0	**<0.001**
Partial	43	33.3	8	6.2	
None	44	34.1	121	93.8	
**Group-II**					
Complete	10	40.0	0	0.0	**<0.001**
Partial	7	28.0	3	12.0	
None	8	32.0	22	88.0	
**Group-III**					
Complete	4	25.0	0	0.0	**0.00** **3**
Partial	6	37.5	1	6.3	
None	6	37.5	15	93.7	
**Group-IV**					
Complete	4	57.1	0	0.0	0.070
Partial	0	0.0	1	14.3	
None	3	42.9	6	85.7	

* Fisher exact test. Statistically significant differences are highlighted in bold.

**Table 6 cancers-17-01388-t006:** Factors associated with overall survival (Cox regression model).

Factor	N	Hazard Ratio	95% CI	*p* Value
Older age at treatment start	231	1.04	1.01–1.06	**0.004**
Higher MCV at baseline	237	0.99	0.98–1.01	0.227
Higher WBC at baseline	237	1.06	1.00–1.12	0.053
Higher BM blasts (%) at baseline	237	1.13	1.09–1.18	**<0.001**
Shorter interval from diagnosis to treatment start	237	0.97	0.96–0.98	**<0.001**
Worse WHO-defined performance status	237	1.28	1.03–1.60	**0.029**
Type of response	231			
Major	141	1.00		
Minor	30	2.35	1.31–4.22	**0.004**
None	41	4.13	2.53–6.73	**<0.001**
Not evaluable	19	6.91	3.70–12.93	**<0.001**
Transfusion dependence	231			
None	72	1.00		
Partial	65	1.37	0.86–2.19	0.118
Complete	94	1.63	1.02–2.60	**0.039**

Abbreviations: BM, bone marrow; CI, confidence interval; MCV, mean corpuscular volume; WBC, white blood cells; WHO, World Health Organization. Statistically significant differences are highlighted in bold.

## Data Availability

Raw data exist in the Greek National Registry EAKMYS and the owner is the Hellenic (Greek) MDS Study Group of the Hellenic Society of Hematology.

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
