# Peer review of "Lenalidomide Efficacy in Patients with MDS and Del-5q: Real-World Data from the Hellenic (Greek) National Myelodysplastic & Hypoplastic Syndromes Registry (EAKMYS)"

_cancers, 2025, doi:10.3390/cancers17091388_

Round 1

Reviewer 1 Report

Comments and Suggestions for Authors

The authors present here a study of efficacy of lenalidomide in MDS patients with 5q- from greek database. There is existing literature that proved the efficacy of lenalidomide in MDS patients with 5q- and also 5q- with one additional cytogenetic abnormality, including cytogenetic response (partial and complete). It is proven that MDS patients with 5q- and multiple other cytogenetic abnormalities have poor prognosis. The article is well written and add to the body of evidence on efficacy of lenalidomide in MDS patients with 5q-. However I have some comments as noted below.

  1. authors should mention how many patients in each of the groups have achieved transfusion independence post treatment with Lenalidomide
  2. Authors should include what was the baseline line hemoglobin levels and what were the hemoglobin levels post lenalidomide treatment.
  3. Authors should define what is major response and what is minor response.
  4. The N in group III is too small to be statistically significant to conclude that lenalidomide is effective in patients with MDS with 5q- and one additional cytogenetic abnormality.

Author Response

Answers to the comments of the first reviewer

Comment 1: Authors should mention how many patients in each of the groups have achieved transfusion independence post treatment with Lenalidomide.

Answer: Thank you for this comment. We have added a new paragraph (3.6) in which we report transfusion needs before and after lenalidomide treatment. We have also added a new Table (Table 5) in which the evolution of transfusion dependence following lenalidomide is shown and comparison between the pre-treatment and post-treatment status has been performed.

Comment 2: Authors should include what was the baseline line hemoglobin levels and what were the hemoglobin levels post lenalidomide treatment.

Answer: We would like to thank again the reviewer for this constructive comment. We have added a new paragraph inside paragraph 3.5 describing the changes of all the hematological parameters after lenalidomide treatment and we have also added a new Table (Table 4), demonstrating these parameters before and after treatment, as well as the results of their comparison.

Comment 3: Authors should define what is major response and what is minor response.

Answer: We would like to thank the reviewer once more. The evaluation of response has been performed according to revised IWG (2018) criteria for response to treatment of MDS patients. This clarification has been added in the Patients and Methods section and the relevant reference has been added.

Comment 4: The N in group III is too small to be statistically significant to conclude that lenalidomide is effective in patients with MDS with 5q- and one additional cytogenetic abnormality.

Answer: The reviewer is right. Only 26 patients with del(5)q plus one additional cytogenetic abnormality have been treated with lenalidomide. Yet this is the largest similar group analyzed and ever reported in one study and the same is true for Group-IV. We mention this issue in the Discussion and in the Limitations of the study. We are planning to repeat our analysis when more patients will have been added, and to potentially perform a meta-analysis of the published data on similar patients.

Argiris Symeonidis

Submission Date of the initial manuscript 28 February 2025

Date of this review 14 Mar 2025 19:11:49

Date of Submission of the revised manuscript and of answering the comments 29 March 2025

Reviewer 2 Report

Comments and Suggestions for Authors

The authors present a research on efficacy of Lenalidomide in MDS patients. The authors have properly designed their experiments and patients data very carefully and properly arranged in groups. All the data is very clean and robust and the manuscript can be recommended for publication with small corrections. 

  1. a negative and positive controls of known drugs should be included.
  2. figure 2, the data can be properly quantified and presented with statistical significance. 

Author Response

Answers to the comments of the second reviewer

Comment 1: A negative and positive controls of known drugs should be included.

Answer: Patients with lower risk MDS and del(5)q do not have treatment options other than Erythropoietin Stimulating Agents and lenalidomide. Other novel treatment approaches for patients with lower risk MDS, such as luspatercept, roxadustat and imetelstat have not yet been applied to this patient population. Thus, the only potentially useful control group would be a group of such patients treated with ESAs. The aim of this retrospective analysis was to investigate mainly efficacy information of lenalidomide on patients with del(5)q. We therefore, have not investigated del(5)q patients treated with ESAs, since this was a clinical practice applied before the approval of lenalidomide for these patients , i.e. before 2007. However, such control groups have been described in the medical literature, and they are mentioned in the relevant references No 42, 43 and 55.

Comment 2: figure 2, the data can be properly quantified and presented with statistical significance. 

Answer: Thank you very much for this comment. We have performed an updated survival analysis in the whole patient group and the updated Kaplan-Meier curves now include information of statistical significance.

Argiris Symeonidis

Submission Date of the initial manuscript 28 February 2025

Date of this review 14 Mar 2025 19:11:49

Date of Submission of the revised manuscript and of answering the comments 29 March 2025

Reviewer 3 Report

Comments and Suggestions for Authors

The paper titled "Lenalidomide Efficacy in Patients with MDS and Del-5q: Real-World Data from the Hellenic (Greek) National Myelodysplastic & Hypoplastic Syndromes Registry (EAKMYS)" presents important clinical findings regarding lenalidomide’s efficacy in treating patients with del(5q) syndrome. It would be beneficial to elaborate on the mechanism of action of lenalidomide, especially regarding how it affects cells or biological systems. This could involve discussing the specific biochemical pathways that are targeted by lenalidomide, its interaction with immune cells, and its therapeutic effects. Expanding on these mechanisms will provide a clearer understanding of the underlying biological processes and the rationale for the treatment's effectiveness in this patient group.

The content between lines 105 and 121 requires proper citations to support the claims or findings presented. It is essential to follow a consistent and clear citation format to avoid confusion. For instance, instead of using “del(5)” (which could be mistaken for a reference number), you can use a more descriptive term like "deletion of the chromosome 5 region" or "chromosome 5 deletion (del(5q))". Additionally, relevant references should be inserted to ensure scientific accuracy and credibility.

Please ensure that the ethics approval number is clearly stated in the manuscript. Ethical review and approval are important for studies involving human or animal subjects, and providing the ethics approval number, along with any relevant institutional review board details, will demonstrate compliance with ethical standards and promote transparency in the research.

Comments on the Quality of English Language

Good

Author Response

It would be beneficial to elaborate on the mechanism of action of lenalidomide, especially regarding how it affects cells or biological systems. This could involve discussing the specific biochemical pathways that are targeted by lenalidomide, its interaction with immune cells, and its therapeutic effects. Expanding on these mechanisms will provide a clearer understanding of the underlying biological processes and the rationale for the treatment's effectiveness in this patient group.

Answer: Thank you for this recommendation, although the mechanism of action of a drug is usually commented, when results of relatively novel drugs are presented, whereas lenalidomide has been approved for MDS-del(5)q, 20 years ago. Anyway, in the second revised version and in the introduction part we have now added the major points of the mechanisms of action of lenalidomide and a whole paragraph is dedicated to this task. 

The content between lines 105 and 121 requires proper citations to support the claims or findings presented.

Answer: In this part we have inserted 12 additional references, indicating the pathophysiological mechanisms, implicated in the pathogenesis of the hematological manifestations of del(5)q MDS and describing their reversal following treatment with lenalidomide.

It is essential to follow a consistent and clear citation format to avoid confusion. For instance, instead of using “del(5)” (which could be mistaken for a reference number), you can use a more descriptive term like "deletion of the chromosome 5 region" or "chromosome 5 deletion (del(5q))".

Answer: We are using the official nomenclature and symbolic system for the description of cytogenetic abnormalities, according to which, the number of the affected chromosome should be placed in brackets, the chromosomal arm, when not identifying the specific area should remain outside the brackets, but when a specific chromosomal area is deleted or translocated, it should be placed in other brackets, adjacent to those including the chromosomal number. This system does not create confusion to the reader with the number of references, because the latter is placed in vertical [ ], but not circular ( ) brackets.  

Additionally, relevant references should be inserted to ensure scientific accuracy and credibility.

Answer: As previously mentioned, we have added 12 additional references, so that the total number of references is now 67.

Please ensure that the ethics approval number is clearly stated in the manuscript. Ethical review and approval are important for studies involving human or animal subjects, and providing the ethics approval number, along with any relevant institutional review board details, will demonstrate compliance with ethical standards and promote transparency in the research.

Answer:  This study represents a harvest of data, which are included in a National Registry. The Hellenic (Greek) National MDS Registry has been inspected and approved by the Greek National Authorities in 2004, and since then several similar retrospective analyses have been performed, the great majority of which have been published in International medical journals. The Registry includes de-identified (anonymized) patient information, and all patients are signing an Informed Consent Form, declaring that they accept their data to be analyzed and presented for scientific purposes by the members of the Hellenic Society of Hematology. Following this, the statisticians are free to perform harvests and analyzed data, according to specific retrospective projects / studies, which are designed and approved by the Steering Committee of the Greek MDS Study Group. For such analyses no additional approval by any Hospital Ethical Committee is required. The same (roughly) regulations are also valid for all National and International Registries.

Reviewer 4 Report

Comments and Suggestions for Authors

This retrospective study examined 238 patients diagnosed with myelodysplastic syndromes (MDS) and del(5q) registered in the Greek National MDS Registry. It assessed the efficacy of lenalidomide across four groups: del(5q) syndrome (Group-I), isolated del(5q) without syndrome (Group-II), del(5q) with one additional cytogenetic abnormality (Group-III), and del(5q) with multiple abnormalities (Group-IV). The major response rates were 78.1%, 58.1%, 50.0%, and 19.0%, respectively, leading to an overall response rate of 81.2%. Importantly, lenalidomide was effective even for patients presenting with excess blasts or AML. These results indicate that lenalidomide may have a wider application than just for isolated del(5q), which could affect treatment strategies and emphasize the necessity for earlier intervention, thus significantly impacting MDS management. However, I have several comments and concerns regarding the current version of the manuscript. I urge the authors to address each comment to revise the manuscript before publication.

Comments for authors

Comment 1: The study suggests lenalidomide's efficacy relates to del(5q) clone dominance. Explain molecular mechanisms (e.g., RPS14 haploinsufficiency, TP53 mutations) supporting this hypothesis and their relevance to patient stratification.

Comment 2: How can the authors attribute outcomes solely to lenalidomide without a control group (e.g., patients treated with hypomethylating agents or supportive care)?

Comment 3: The additional cytogenetic abnormalities in Groups III and IV are listed (e.g., trisomy 8, del(20q)), but their specific impact on response is unclear. Provide a subgroup analysis to assess their individual prognostic significance.

Comment 4: With small sample sizes in Groups III (26) and IV (25), are the statistical analyses sufficiently powered to detect differences?

Comment 5: Provide a detailed adverse event profile to assess tolerability comprehensively.

Comment 6: I encourage the authors to proofread and correct any grammatical issues carefully to enhance readability.

Author Response

Answers to the comments of the fourth reviewer

Comment 1: The study suggests lenalidomide's efficacy relates to del(5q) clone dominance. Explain molecular mechanisms (e.g., RPS14 haploinsufficiency, TP53 mutations) supporting this hypothesis and their relevance to patient stratification.

Answer: This is a thoughtful comment that should be valid for any prospective study. In our study we have not used molecular criteria for patient stratification, since this is a retrospective chart review efficacy analysis, of patients registered in a National Registry. Molecular data have started to be tested and incorporated in the Registry the recent years, and the majority of included patients have not undergone molecular analysis, since they have been diagnosed and treated before 2017, when the importance of NGS analysis has started to be tested routinely in MDS. Thus, we have stratified our patients, using clear cytogenetic and disease classification criteria.

Comment 2: How can the authors attribute outcomes solely to lenalidomide without a control group (e.g., patients treated with hypomethylating agents or supportive care)?

Answer: Thanks for this comment. However, we have to stress your attention that this study represents an analysis of real-world data of lenalidomide use in patients with lower risk MDS. In our study we have only analyzed patients exhibiting del(5)q alone or with additional cytogenetic abnormalities. Some of these patients had only received ESAs before lenalidomide, and they have been treated according to the National and International guidelines, and according to the treating physician opinion. Thus, no specific design for a prospective study is available.     

Comment 3: The additional cytogenetic abnormalities in Groups III and IV are listed (e.g., trisomy 8, del(20q)), but their specific impact on response is unclear. Provide a subgroup analysis to assess their individual prognostic significance.

Answer: Thanks again for this recommendation. Indeed, if the number of patients with additional cytogenetic abnormalities was higher, we would have tried to perform subgroup analysis according to specific abnormalities. However, with data on 25 patients only we were unable to perform any subgroup analysis. Yet, this is a definite future task, when the experience of the treatment of such patients will be increased.

Comment 4: With small sample sizes in Groups III (26) and IV (25), are the statistical analyses sufficiently powered to detect differences? 

Answer: We agree with the reviewer that the number of patients of Groups-III and -IV are not high enough to draw powered analyses for potential treatment recommendation, as we mentioned in the previous answer. However please, permit us to underline, that these are the highest number of such patients ever, homogeneously analyzed in one study. We have not encountered in the available to us, medical literature, publications analyzing higher number of patients with MDS-Del(5)q plus one or more than one additional cytogenetic abnormalities. The number of patients analyzed in our study represents a harvest of 17 years of activity, following lenalidomide approval for MDS-Del(5)q. We are sorry for not having analyzed additional patients, we may consider in the future to perform a meta-analysis of scarcely reported similar patients and of case reports, to potentially draw clearer results.   

Comment 5: Provide a detailed adverse event profile to assess tolerability comprehensively.

Answer: In paragraph 3.9 we report in detail the adverse events, which we have detected in the treated patient population as a whole, and in each of the four specified patient groups. In general, we did not observe any unexpected AEs, and not surprisingly, neutropenia and thrombocytopenia were more commonly encountered among patients with an excess of marrow blasts and in those with more than one additional cytogenetic abnormalities. 

Comment 6: I encourage the authors to proofread and correct any grammatical issues carefully to enhance readability.

Answer: Thank you for this recommendation. We have performed thoroughly proofreading of the final manuscript and we have asked the opinion of an expert in English language, to read and potentially improve/correct all linguistic issues. We have been reassured that the manuscript is written well, and no additional intervention is required.

Argiris Symeonidis                                                                                                                                                      Corresponding Author

Submission Date of the initial manuscript 28 February 2025                                                                                    Date of this review 14 Mar 2025 19:11:49                                                                                                                    Date of Submission of the revised manuscript and of answering the comments 29 March 2025

Round 2

Reviewer 1 Report

Comments and Suggestions for Authors

The authors have made the revisions satisfactorily. recommend publication

Reviewer 4 Report

Comments and Suggestions for Authors

The revised manuscript addresses my comments and concerns effectively. I believe the manuscript is now acceptable.